# LIMITS OF ALGORITHMIC STABILITY FOR DISTRIBUTIONAL GENERALIZATION

## ABSTRACT

As machine learning models become widely considered in safety critical settings, it is important to understand when models may fail after deployment. One cause of model failure is distribution shift, where the training and test data distributions differ. In this paper we investigate the benefits of training models using methods which are algorithmically stable towards improving model robustness, motivated by recent theoretical developments which show a connection between the two. We use techniques from differentially private stochastic gradient descent (DP-SGD) to control the level of algorithmic stability during training. We compare the performance of algorithmically stable training procedures to stochastic gradient descent (SGD) across a variety of possible distribution shifts - specifically covariate, label, and subpopulation shifts. We find that models trained with algorithmically stable procedures result in models with consistently lower generalization gap across various types of shifts and shift severities as well as a higher absolute test performance in label shift. Finally, we demonstrate that there is there is a tradeoff between distributional robustness, stability, and performance.

## 1 INTRODUCTION

As machine learning (ML) is applied in several high-stakes decision making situations such as healthcare (Ghassemi et al., 2017; Rajkomar et al., 2018; Zhang et al., 2021a) and lending (Liu et al., 2018; Weber et al., 2020), it is important to consider scenarios when models fail. Typically, models are trained with empirical risk minimization (ERM), which assumes that the training and test data are sampled i.i.d from the same underlying distribution (Vapnik, 1999). Unfortunately, this assumption means that ERM is susceptible to performance degradation under distribution shift (Nagarajan et al., 2021). Distribution shift occurs when the data distribution encountered during deployment is different, or changes over time while the model is used. In practice, even subtle shifts can significantly affect model performance (Rabanser et al., 2019). Given that distribution shift is a significant source of model failure, there has been much work directed toward improving model robustness to distribution shifts (Taori et al., 2020; Cohen et al., 2019; Engstrom et al., 2019; Geirhos et al., 2018; Zhang et al., 2019; Zhang, 2019).

One concept recently introduced to improve model robustness is *distributional generalization* (Kulynych et al., 2022; Nakkiran & Bansal, 2020; Kulynych et al., 2020). Distributional generalization (DG) extends classical generalization to encompass any evaluation function (instead of just the loss objective) and allows the train and test distributions to differ. Kulynych et al. (2022) prove that algorithms which satisfy total variation stability (TV stability) bound the gap between train and test metrics when distribution shift is present, i.e., algorithms which satisfy TV stability are also satisfy DG. This motivates the use of techniques from differentially private (DP) learning to satisfy DG, since DP implies TV stability (Kulynych et al., 2022). We know from other works that DP learning often comes at a cost to accuracy (Bagdasaryan et al., 2019; Suriyakumar et al., 2021; Jayaraman & Evans, 2019). Unfortunately these works don't thoroughly explore the empirical implications of their theorems across a wide variety of settings except for a positive result in Suriyakumar et al. (2021). Because robustness to new settings is an important question for deployments of models, it is important to understand how the theory of distributional robustness will work practically when facing different types and severities of shifts. Furthermore, it is hard to understand from the current theory how practitioners should tune the level of stability as to achieve high performing models.

In this paper we conduct an extensive empirical study on the impact of using algorithmically stable learning strategies for robustness when facing distribution shift. Stable learning (SL) refers to approaches that constrain the model optimization objective or learning algorithm to improve model stability. We focus on two questions regarding the use of SL for DG in practice: **(i)** Under what types of shift is SL more robust and accurate than ERM? **(ii)** Are SL trained models consistently robust across all hyperparameters, model architectures, and shift severities? We target four common examples of shift: covariate (Shimodaira, 2000), label (Lipton et al., 2018; Storkey, 2009), subpopulation (Duchi & Namkoong, 2021; Koh et al., 2021), and natural shifts Taori et al. (2020). We use state of the art models and large benchmark datasets focusing on realistic prediction tasks in object recognition, satellite imaging, biomedical imaging, and clinical notes (see Table 1, with details in Section 4.2). The primary comparison we make is through the generalization gap, defined as the difference in model performance between training and testing (Zhang et al., 2021b). Under extensive experimentation–incorporating 32 distinct types of distribution shift and 5 severity levels–we find:

1. SL improves both accuracy and robustness for label and natural shifts.
2. SL has a robustness-accuracy tradeoff for covariate and subpopulation shift.
3. The tradeoffs of SL are consistent across different shift severities, model architectures, and hyperparameter settings.

## 2  RELATED WORK

Many approaches have been developed in pursuit of robustness to distribution shift, including: domain adaption (Wang & Deng, 2018), out-of-distribution detection (Yang et al., 2021), adversarial training (Madry et al., 2018; Ilyas et al., 2019), as well as through algorithmic improvements (Sagawa et al., 2019). To solve the distribution shift problem, many recent techiques for distributionally robust optimization (DRO), such as risk averse learning (Curi et al., 2020), have been developed. However, many of these methods do not perform better than ERM (Pfohl et al., 2022) and involve complex implementations, making them difficult to use.

Algorithmic stability has also been explored to improve distributional robustness. It is often easier to implement, with simpler methods such as: $\ell_2$ regularization (Wibisono et al., 2009), early stopping (Hardt et al., 2016), and differentially private stochastic gradient descent (DP-SGD) (Abadi et al., 2016). Early stopping and $\ell_2$ regularization have already been studied for their potential to improve distributional robustness (Sagawa et al., 2019). However, it's difficult to conduct fine-grained analyses into improved robustness with these methods because their stability is not directly controllable. This motivates our use of DP-SGD to investigate the limits of stability for DG since we can control the level of stability by adjusting the noise multiplier $\sigma$ in DP-SGD. While algorithmic stability has been explored theoretically in previous works (see Section 3), we explore it empirically in this paper across various synthetic and natural distribution shifts.

## 3  BACKGROUND AND NOTATION

We provide an overview of the connections between algorithmic stability, DP, and different forms of generalization in this section. It is well-established that algorithmic stability implies generalization in the traditional ERM setup (Bousquet & Elisseeff, 2002). Additional work has proven that DP implies stability and thus, implies generalization (Bassily et al., 2016; Dwork et al., 2015). In this section we define these concepts and draw connections between them. This is done to clarify the theoretical implication that DP leads to improved distributional robustness.

**Notation**  We assume there is a training dataset $D_{train} = \{(x_i, y_i)\}_{i=1}^n$ of labeled examples such that $D_{train} \sim \mathcal{D}$ and a testing dataset $D_{test} = \{(x_i, y_i)\}_{i=1}^m$ of labeled examples such that $D_{test} \sim \mathcal{D}'$. Given $D_{train}$, we use a randomized learning algorithm $\mathcal{M}(D_{train})$ to learn parameters $\theta \in \Theta$ of a model relating the datapoints $\{x_i\}$ to their corresponding label $\{y_i\}$.

Now we will describe differential privacy and its links to stability.

**Definition 1** (Differential Privacy (Dwork et al., 2006))**.** *Suppose we have two datasets $D, D'$ which have a Hamming distance (the number of examples which the two databases differ by) of 1, then an algorithm $\mathcal{M}(D)$ is $(\epsilon, \delta)$-differentially private if:*

$$Pr[\mathcal{M}(D) \in \Theta] \leq exp(\varepsilon)Pr[\mathcal{M}(D') \in \Theta] + \delta \tag{1}$$

*where* $\Theta \subseteq Range(\mathcal{M})$.

By definition, DP guarantees privacy by bounding the effect that any individual datapoint has on the output of $\mathcal{M}$. This leads to DP implying strong forms of stability such as TV stability (Bassily et al., 2016) and uniform stability (Wang et al., 2016). Next, we will define these notions of algorithmic stability and show how DP implies them both.

**Definition 2** (TV Stability). *Suppose we have two datasets $D, D'$ which have a Hamming distance of 1, then an algorithm $\mathcal{M}(D)$ is $(\delta)$-TV stable if:*

$$Pr[\mathcal{M}(D) \in \Theta] \leq Pr[\mathcal{M}(D') \in \Theta] + \delta \tag{2}$$

We can also write this as the an upper bound on the total variation distance between two distributions $P$ and $Q$ where $D \sim P$ and $D' \sim Q$ between $\mathcal{M}(D)$ and $\mathcal{M}(D')$. Where $d_{TV}(P, Q) = \sup_T |P(T) - Q(T)|$, then TV stability is $d_{TV}(\mathcal{D}, \mathcal{D}') \leq \delta$.

**Definition 3** (Uniform Stability (Bousquet & Elisseeff, 2002)). *Suppose we have two datasets $D, D'$ which have a Hamming distance of 1, then an algorithm $M(D)$ is $(\delta)$-uniformly stable if:*

$$\underset{s.\,t.\|D-D'\|=1}{\forall D, D'} |\mathbb{E}\ell(D; \mathcal{M}(D)) - \mathbb{E}\ell(D; \mathcal{M}(D'))| \leq \delta \tag{3}$$

If the loss function $\ell$ is bounded between [0,1] then TV stability implies $\delta$-uniform stability (Kulynych et al., 2022). All of these definitions assume we are sampling $D$ and $D'$ from the same underlying distribution. When distribution shift ocurrs, this is no longer true. Thus, we will present the results of Kulynych et al. (2022) who demonstrate that TV-stable algorithms satisfy a notion of generalization that captures distribution shift known as *distributional generalization*.

**Definition 4** (Distributional Generalization (Kulynych et al., 2022; Nakkiran & Bansal, 2020; Kulynych et al., 2020)). *Given two datasets $D$ and $D'$ sampled from two different distributions $P$ and $Q$, an algorithm $\mathcal{M}(D)$ satisfies $\delta$-distributional generalization (DG) if for all $\phi: D \times \Theta \to [0, 1]$*

$$|\underset{D \sim P}{\mathbb{E}}\phi(D; \mathcal{M}(D)) - \underset{D' \sim P, D' \sim Q}{\mathbb{E}}\phi(D'; \mathcal{M}(D))| \leq \delta \tag{4}$$

Kulynych et al. (2022) prove that any algorithm which is $\delta$-TV stable is $\delta$-DG. Thus implying that algorithmic stability improves robustness to distribution shift. The level of TV stability and DG which are parameterized by $\delta$ are directly correlated with level of noise $\sigma$ we use in DP-SGD. Throughout the rest of the paper, the larger $\sigma$ is the more stable the algorithm is (i.e. the lower $\delta$ is).

## 4 METHODS

To better understand the potential and limits for using algorithmic stability to improve model robustness we conduct a thorough empirical study across several datasets, types of shifts, and shift severities. We explore both synthetic and natural distribution shifts that arise due to differences in the covariate, label, and subpopulation distributions between training and testing (Table 1). Our empirical investigation covers more than 200 experiments, incorporating 32 distinct forms of distribution shift with varying levels of severity.

In each of our experiments—given a training dataset $\mathcal{D}_{\text{train}}$ and test dataset $\mathcal{D}_{\text{test}}$ with known distribution shift—we compare the difference in generalization gap (Definition 5) of models with and without algorithmic stability ( "stable learning" (SL)) and ERM respectively. Characterizing this gap is an important step to determine how well the theoretical guarantees of DG hold in practice.

We aim to provide answers to the following critical questions about practical use of SL for DG, motivating its use beyond theory: **(i)** Under what types of shift is SL more robust and accurate than ERM? **(ii)** Are SL trained models consistently robust across all hyperparameters, model architectures, and shift severities?

| Dataset | Shift Type | Size of Dataset | Models | Prediction Task | Performance Metric |
|---------|------------|-----------------|--------|-----------------|--------------------|
| CIFAR10-C | Synthetic Covariate | 105,000 | end-to-end CNN | Classification | Accuracy |
| Imbalanced CIFAR10 | Synthetic Label | 115,000 | end-to-end CNN | Classification | Accuracy |
| Waterbirds | Synthetic Subpopulation | 4,795 | Logistic Regression | Classification | Worst-Group Accuracy |
| COOS | Natural Covariate | 132,209 | end-to-end ResNet18 | Classification | Accuracy |
| PovertyMap | Natural Subpopulation & Label | 19,669 | end-to-end ResNet18 | Regression | Pearson Corr Coefficient |
| MIMIC-III | Natural Subpopulation & Label | 25,879 | Logistic Regression | Mortality | Worst-Group AUC |

Table 1: The datasets, prediction tasks, and model architectures used throughout the paper to evaluate the relationship between algorithmic stability and distribution shift.

## 4.1 Distribution Shift

We present and define the three different kinds of distribution shifts that we explore in our empirical study.

Consider a joint distribution $\mathcal{D}$ between features $X$ and labels $Y$, $\mathcal{D} = X \times Y$, with training data $\mathcal{D}_{\text{train}}$ and testing data $\mathcal{D}_{\text{test}}$. We define *distribution shift* to be any change to the marginal, conditional, or joint distributions of $X$ and $Y$ between $\mathcal{D}_{\text{train}}$ and $\mathcal{D}_{\text{test}}$, based on prior work (Quionero-Candela et al., 2009; Rabanser et al., 2019; Duchi & Namkoong, 2018). We explain the three types of shift below:

**Covariate shift** occurs when the features $X$ change between training and testing, but the prediction given a feature stays the same, such that $P(X_{\text{train}}) \neq P(X_{\text{test}})$, $P(Y_{\text{train}}|X_{\text{train}}) = P(Y_{test}|X_{test})$. A natural example of covariate shift arises when datasets of MRI images are recorded by different machines, providing different features but not changing disease prevalence.

**Label shift** is when the distribution of labels changes, but the class-conditional densities are constant, such that $P(Y_{\text{train}}) \neq P(Y_{\text{test}})$, $P(X_{\text{train}}|Y_{\text{train}}) = P(X_{\text{test}}|Y_{\text{test}})$. This shift occurs when one class is overrepresented in the dataset, but the underlying feature distribution is conditionally equivalent.

**Subpopulation shift** is when the distribution of subpopulations (e.g. subgroups defined by the intersection of sex and race) changes. Define $G$ to be a discrete random variable representing membership in a subpopulation. Subpopulation shift occurs when $\exists g \in G$ s.t. $P(X_{train}, Y_{train}|G = g) \neq P(X_{test}, Y_{test}|G = g)$. For example, when we move between rural and urban medical centers the proportion of subpopulations changes where minority populations are less represented in rural settings.

**Natural shift** are those where the training and test distributions are different from real-world data collection, without additional manipulation of data. These natural shifts can be combinations of covariate, label, subpopulation, and other types of shifts.

## 4.2 Datasets

We investigate synthetic covariate, label, and subpopulation shifts, as well as naturally occurring instances of covariate and subpopulation shifts. We use the definition of natural and synthetic shift from Taori et al. (2020). Synthetic shifts are those where the data originally is all from the same distribution but is manipulated such that the training and test distributions are different. Meanwhile, natural shifts are those where the training and test distributions are already different.

**Synthetic datasets:** For covariate shift, we use the `CIFAR10-C` dataset, created by shifting CIFAR10's test set with 19 types of algorithmically generated corruptions of 5 different severities for a total of 95 different synthetic covariate shifts (Hendrycks & Dietterich, 2019). For synthetic label shift, we use `Imbalanced-CIFAR10`, which we created by inducing a class imbalance in CIFAR10 to create a shift in $P(Y)$. These shifts were created randomly, where the percentage of of samples in the shifted dataset from the original test dataset was was chosen randomly from $10 - 100\%$. To explore synthetic subpopulation shifts, we use the `Waterbirds` dataset (Sagawa et al., 2019) made up of bird images with synthetic backgrounds.

**Natural datasets:** Our natural covariate shifts are derived from the Cells-Out-of-Sample (COOS) dataset, which consists of mouse cell images of 7 biological classes, with 4 separate test sets of increasing degrees of covariate shift (Lu et al., 2019). We also explore natural subpopulation and label shifts with the PovertyMap dataset predicting poverty levels from satellite imagery (Koh et al., 2021), and MIMIC-III clinical notes predicting mortality (Johnson et al., 2016).

For more detailed information about the datasets used in this paper, please refer to Appendix A.2.

### 4.3 MODEL TRAINING

We train models using DP-SGD (with varying levels of noise and clipping as a way to modify the amount of stability, as detailed in Appendix A.1.2 and Appendix A.3) and ERM for each dataset and type of shift. Each model is trained with early stopping on the validation loss to prevent overfitting. The scale of the hyperparameter search and noise levels we used for determining the best performing models can be found in Appendix A.3. For all experiments, we use the Opacus package (Yousefpour et al., 2021) or Tensorflow Privacy package (McMahan & Andrew, 2018) to implement DP-SGD. Models and data references are given in Table 1. Note that the models used for the CIFAR dataset (Tramèr & Boneh, 2021) in covariate and label shift are small models created for differential privacy, to mitigate the dimensional dependence of DP-SGD Yu et al. (2017). Thus, the reported ERM test accuracy is lower than state-of-the-art performance using larger model architectures such as the ResNet (He et al., 2016) on the same dataset. When we compare against distributionally robust optimization (DRO) results, we use the conditional value at risk (CVaR) optimization algorithm (Lévy et al., 2020).

### 4.4 EXPERIMENTAL PROCEDURE

For each pair of $\mathcal{D}_{\text{train}}$ and $\mathcal{D}_{\text{test}}$ in an experiment, we perform the following procedure:

- Train a set of models $\{M_1, M_2, ...\}$ (performing a hyperparameter search) on $\mathcal{D}_{\text{train}}$ with DP-SGD using different noise multiplier values $\{\sigma_1, \sigma_2, ...\}$ as varying levels of algorithmic stability. We also train ERM (not stable) and DRO (another stable algorithm) models as comparative baselines. SL models are trained with DP-SGD, detailed in Appendix A.1.2.
- Test each model in the sets $\{M_1, M_2, ...\}$ on shifted testing datasets $\{\mathcal{D}'_1, \mathcal{D}'_2...\}$. While covariate and subpopulation shift have shifted test datasets, label shift has a shifted training dataset, as practitioners would see in a scenario of class-imbalanced training data.
- Measure the generalization gap $G$, the difference between training and testing accuracy for SL and ERM models, $G_{SL}$ and $G_{ERM}$. To measure the improvement SL has over ERM, we report the difference in generalization gap $\Delta_G$ as defined in Definition 5 below.

### 4.5 EVALUATION

We design our empirical investigation to answer the two previous questions posed at the outset of Section 4 on the limits of using SL for distributional robustness. We address them as follows:

**(i)** The metric we use to primarily compare the robustness of models is the difference in generalization gap, $\Delta_G$ (Definition 5). If $\Delta_G > 0$, this indicates that the model trained with a stable learning algorithm has a lower generalization gap and is therefore more robust than ERM trained models. As mentioned in Section 4.4, we test across various levels of stability to find the optimal $\sigma$ value.

**Definition 5** (Difference in Generalization Gap). *Given the training and $D$ and $D'$ sampled from two different distributions $P$ and $Q$, an ERM model $\mathcal{M}_{ERM}(D)$ and alternate training algorithm $A$ with model $\mathcal{M}_A(D)$ and metric $\phi: D \times \Theta \rightarrow \mathcal{R}$ we define the generalization gap as:*

$$\Delta_G = \left| \underset{D \sim P}{\mathbb{E}} \phi(D; \mathcal{M}_{ERM}(D)) - \underset{D \sim P, D' \sim Q}{\mathbb{E}} \phi(D'; \mathcal{M}_{ERM}(D)) \right| - \left| \underset{D \sim P}{\mathbb{E}} \phi(D; \mathcal{M}_A(D)) - \underset{D \sim P, D' \sim Q}{\mathbb{E}} \phi(D'; \mathcal{M}_A(D)) \right|$$

(5)

We calculate $\Delta_G$ for SL/DRO compared to ERM , referred to SL $\Delta_G$/DRO $\Delta_G$.

**(ii)** We investigate if stability holds over different shift severities using the synthetic datasets by evaluating for which shifts is $\Delta_G > 0$. Shift severity is characterized as the distance between $\mathcal{D}_{\text{train}}$

| Shift Severity | ERM train acc | ERM test acc | SL train acc | SL test acc | DRO train acc | DRO test acc | SL Accuracy Gain | SL $\Delta_G$ | DRO Accuracy Gain | DRO $\Delta_G$ |
|---|---|---|---|---|---|---|---|---|---|---|
| None | $0.938 \pm 0.001$ | $0.771 \pm 0.034$ | $0.748 \pm 0.165$ | $0.700 \pm 0.113$ | $0.770 \pm 0.032$ | $-0.071 \pm 0.113$ | $-0.071 \pm 0.113$ | $\mathbf{0.119} \pm 0.057$ | $-0.001 \pm 0.021$ | $0.014 \pm 0.002$ |
| 1 | $0.923 \pm 0.025$ | $0.748 \pm 0.042$ | $0.731 \pm 0.072$ | $0.666 \pm 0.029$ | $0.892 \pm 0.021$ | $0.687 \pm 0.063$ | $-0.082 \pm 0.037$ | $\mathbf{0.109} \pm 0.061$ | $-0.061 \pm 0.051$ | $-0.029 \pm 0.015$ |
| 2 | $0.919 \pm 0.000$ | $0.686 \pm 0.050$ | $0.808 \pm 0.089$ | $0.646 \pm 0.094$ | $0.892 \pm 0.015$ | $0.676 \pm 0.073$ | $-0.041 \pm 0.057$ | $\mathbf{0.071} \pm 0.047$ | $-0.01 \pm 0.045$ | $0.016 \pm 0.002$ |
| 3 | $0.892 \pm 0.049$ | $0.649 \pm 0.097$ | $0.780 \pm 0.139$ | $0.612 \pm 0.124$ | $0.895 \pm 0.012$ | $0.588 \pm 0.094$ | $-0.038 \pm 0.046$ | $\mathbf{0.075} \pm 0.127$ | $-0.061 \pm 0.095$ | $-0.064 \pm 0.021$ |
| 4 | $0.875 \pm 0.043$ | $0.618 \pm 0.067$ | $0.783 \pm 0.120$ | $0.556 \pm 0.104$ | $0.886 \pm 0.033$ | $0.584 \pm 0.075$ | $-0.060 \pm 0.059$ | $\mathbf{0.032} \pm 0.096$ | $-0.034 \pm 0.075$ | $-0.046 \pm 0.020$ |
| 5 | $0.857 \pm 0.046$ | $0.538 \pm 0.115$ | $0.664 \pm 0.144$ | $0.468 \pm 0.124$ | $0.879 \pm 0.036$ | $0.484 \pm 0.118$ | $-0.070 \pm 0.045$ | $\mathbf{0.122} \pm 0.111$ | $-0.054 \pm 0.116$ | $-0.075 \pm 0.023$ |

Table 2: SL demonstrates an accuracy-robustness tradeoff in the `CIFAR-C` dataset representing synthetic covariate shifts. We observe that for all shifts, SL $\Delta_G > 0$, indicating that each SL model is more robust than ERM. However, there is a consistent loss in accuracy for both SL. We present results across five shift severities for $\sigma = 0.1$. We observe a similar tradeoff for the DRO CVaR algorithm, indicating that these algorithms cannot improve both robustness and accuracy with covariate shift.

| Shift Severity | ERM train acc | ERM test acc | SL train acc | SL test acc | Accuracy Gain | $\Delta_G$ |
|---|---|---|---|---|---|---|
| None | $0.913 \pm 0.002$ | $0.864 \pm 0.020$ | $0.852 \pm 0.093$ | $0.796 \pm 0.123$ | $-0.068 \pm 0.089$ | $\mathbf{0.007} \pm 0.084$ |
| 1 | $0.817 \pm 0.202$ | $0.778 \pm 0.232$ | $0.790 \pm 0.215$ | $0.748 \pm 0.223$ | $-0.031 \pm 0.009$ | $-\mathbf{0.004} \pm 0.022$ |
| 2 | $0.900 \pm 0.267$ | $0.777 \pm 0.282$ | $0.812 \pm 0.120$ | $0.749 \pm 0.161$ | $-0.028 \pm 0.121$ | $\mathbf{0.054} \pm 0.025$ |
| 3 | $0.897 \pm 0.250$ | $0.773 \pm 0.251$ | $0.812 \pm 0.109$ | $0.742 \pm 0.157$ | $-0.031 \pm 0.094$ | $\mathbf{0.054} \pm 0.048$ |
| 4 | $0.870 \pm 0.197$ | $0.715 \pm 0.207$ | $0.813 \pm 0.134$ | $0.7331 \pm 0.155$ | $0.018 \pm 0.052$ | $\mathbf{0.074} \pm 0.011$ |
| 5 | $0.937 \pm 0.265$ | $0.765 \pm 0.200$ | $0.850 \pm 0.000$ | $0.731 \pm 0.160$ | $-0.034 \pm 0.041$ | $\mathbf{0.053} \pm 0.154$ |

Table 3: SL demonstrates an accuracy-robustness tradeoff in the `Waterbirds` datasets representing synthetic subpopulation shifts. We observe that for all shifts, $\Delta_G > 0$, indicating that each SL model is more robust than ERM. However, there is a consistent loss in accuracy. We present results across five shift severities for $\sigma = 0.1$.

and $\mathcal{D}_{\text{test}}$. We use the Optimal Transport Dataset Distance (OTDD) (Alvarez-Melis & Fusi, 2020). We choose this metric as opposed to other dataset distances for its provable guarantees and that it allows for completely disjoint datasets to be compared. We categorize our synthetic datasets by shift severity by first normalizing the computed OTDD and sorting them into quintiles 1-5.

**(iii)** To explore the tradeoff between robustness and accuracy, we also report model performance throughout the paper and compare it to $\Delta_G$. Model performance is reported as accuracy except for `MIMIC-III` and `PovertyMap`, where area under the curve (AUC) and Pearson Correlation Coefficient are used, respectively, due to change in classification task (See Table 1). In `MIMIC-III`, related work focuses on AUC because it is the standard metric used for diagnostics. Thus, we use AUC since it is the standard for evaluating clinical prediction models for mortality. In `PovertyMap`, predicts a real-valued composite asset wealth index from satellite images, and thus, the models are evaluated on the Pearson correlation ($r$) between their predicted and actual asset wealth indices.

## 5 STABILITY HAS POOR ROBUSTNESS-ACCURACY TRADEOFFS FOR COVARIATE AND SUBPOPULATION SHIFTS

In this section, we examine results from our synthetic covariate and subpopulation experiments seen in Table 2 and Table 3. We investigate potential sources of why this tradeoff exists.

We observe that for all shift severities in covariate `CIFAR-C` and subpopulation `Waterbirds`, SL has increased robustness as compared to ERM, with $\Delta_G > 0$. However, for both shifts there is a tradeoff between robustness and accuracy, seen by the consistent negative accuracy gains, which increases with shift severity.

We also find that SL is more robust on the natural covariate shifted `COOS` but at the expense of accuracy (Table 4). We find that this result holds across all values of stability we tried. This negative finding is not surprising because each of the shifts in `COOS` are covariate shifts.

| Shift Severity | SL train acc | SL test acc | ERM train acc | ERM test acc | Accuracy Gain | $\Delta_G$ |
|---|---|---|---|---|---|---|
| 1 | 0.872 | 0.840 | 0.907 | 0.852 | -0. 012 | **0.022** |
| 2 | 0.872 | 0.843 | 0.907 | 0.853 | -0.010 | **0.024** |
| 3 | 0.872 | 0.810 | 0.907 | 0.706 | **0.166** | **0.033** |
| 4 | 0.872 | 0.467 | 0.907 | 0.559 | -0.092 | **-0.056** |

Table 4: SL is more robust to most natural shifts found in `COOS` except for the most severe. This is most likely due to lower model accuracy. These results are for $\sigma = 0.1$. Standard deviations are not provided because of computational constraints.

To gain insight into the nature of this robustness-accuracy tradeoff, we compare SL to a DRO algorithm that optimizes conditional value at risk (CVaR). Through empirical investigation we uncover that both methods fail on covariate shift (see Table 2). More interestingly, these methods are failing for different reasons. Investigation into the underlying cause has led to the formulation of following conjecture: We believe that SL mimics learning under a transformation of $P(X)$ such that it becomes closer to the uniform distribution as we increase the level of stability.

With enough uniform stability, $P(X)$ would be the Uniform distribution, equivalent to eliminating all signal from the covariates and predicting randomly. Similar to covariate shift, as SL approaches the uniform distribution, information about the changed $P(X|G)$ during subpopulation shift is lost. In contrast, the DRO CVaR optimization focuses on the tails of the distribution Levy et al. (2020) providing limited utility for most covariate/subpopulation shifts which oftentimes apply the same transform to every point in the distribution (Shimodaira, 2000; Duchi & Namkoong, 2018).

These results and the corresponding conjecture lead to the conclusion that SL under uniform stability is not a good candidate for improving robustness under covariate or subpopulation shift, as it comes at a major cost to accuracy.

# 6 STABILITY IMPROVES ROBUSTNESS AND ACCURACY TO LABEL SHIFT AND NATURAL SHIFTS

In this section, we demonstrate that stability improves both robustness and accuracy to label shifts and natural shifts. Furthermore, we investigate potential sources of this improvement compared to ERM. We draw on similarities between importance weighting, distributionally robust optimization, and stable learning to help understand these improvements.

We demonstrate these results first on a variety of label shifts on `Imbalanced-CIFAR` (Table 5). Even as the shift severity increases, SL outperforms ERM, with $\Delta_G > 0$ and a positive accuracy gain. However, these improvements occur at a specific level of stability $\sigma = 0.1$ (i.e. the amount of noise in DP-SGD). At stronger levels of stability we find that robustness is better than ERM at the expense of accuracy (Table 11). This is the first result of many throughout our work which indicates that level of stability is a hyperparameter which should be tuned to find the best level of robustness *and* accuracy.

We demonstrate similar improvements when testing against natural distribution shifts. SL improves both robustness and accuracy on `MIMIC-III` and `PovertyMap` (Table 6). Specifically, we see an increase in accuracy of 2.9% and and 11.3% and increase in robustness of 4.2% and 0.8% respectively. Similar to our label shift results we had to tune the level of stability, again supporting the observation that it is a hyperparameter that must be tuned.

We first investigate why stability outperforms ERM on both robustness *and* accuracy on label shifts. We demonstrate that stable learning (especially those that satisfy uniform stability like DP-SGD) mimics training that we would see if our training label distribution was uniform over all labels. We show this by showing similar results when training with DRO (to mimic a uniform label distribution) (Table 5). We show both of these methods also improve accuracy and robustness to a similar degree that SL does.

| Shift Severity | ERM train acc | ERM test acc | SL train acc | SL test acc | DRO train acc | DRO test acc | SL Accuracy Gain | SL $\Delta_G$ | DRO Accuracy Gain | DRO $\Delta_G$ |
|---|---|---|---|---|---|---|---|---|---|---|
| None | $0.919 \pm 0.001$ | $0.771 \pm 0.034$ | $0.748 \pm 0.165$ | $0.700 \pm 0.113$ | $0.904 \pm 0.021$ | $0.770 \pm 0.032$ | $-0.071 \pm 0.113$ | $\mathbf{0.119} \pm 0.057$ | $-0.001 \pm 0.021$ | $0.014 \pm 0.002$ |
| 1 | $0.776 \pm 0.242$ | $0.577 \pm 0.174$ | $0.824 \pm 0.098$ | $0.630 \pm 0.081$ | $0.868 \pm 0.014$ | $0.676 \pm 0.005$ | $\mathbf{0.086} \pm 0.091$ | $\mathbf{0.013} \pm 0.052$ | $\mathbf{0.099} \pm 0.091$ | $\mathbf{0.007} \pm 0.002$ |
| 2 | $0.863 \pm 0.079$ | $0.576 \pm 0.066$ | $0.824 \pm 0.026$ | $0.667 \pm 0.006$ | $0.88 \pm 0.002$ | $0.683 \pm 0.002$ | $\mathbf{0.091} \pm 0.061$ | $\mathbf{0.045} \pm 0.038$ | $\mathbf{0.107} \pm 0.061$ | $\mathbf{0.009} \pm 0.003$ |
| 3 | $0.734 \pm 0.121$ | $0.570 \pm 0.118$ | $0.829 \pm 0.200$ | $0.665 \pm 0.034$ | $0.859 \pm 0.031$ | $0.679 \pm 0.003$ | $\mathbf{0.095} \pm 0.168$ | $\mathbf{0.001} \pm 0.054$ | $\mathbf{0.109} \pm 0.018$ | $\mathbf{-0.01} \pm 0.004$ |
| 4 | $0.682 \pm 0.210$ | $0.509 \pm 0.128$ | $0.819 \pm 0.234$ | $0.700 \pm 0.217$ | $0.864 \pm 0.011$ | $0.695 \pm 0.003$ | $\mathbf{0.191} \pm 0.217$ | $\mathbf{0.064} \pm 0.080$ | $\mathbf{0.186} \pm 0.032$ | $\mathbf{0.004} \pm 0.001$ |
| 5 | $0.590 \pm 0.057$ | $0.417 \pm 0.041$ | $0.819 \pm 0.014$ | $0.670 \pm 0.014$ | $0.852 \pm 0.011$ | $0.684 \pm 0.012$ | $\mathbf{0.252} \pm 0.047$ | $\mathbf{0.024} \pm 0.020$ | $\mathbf{0.267} \pm 0.021$ | $\mathbf{-0.005} \pm 0.001$ |

Table 5: SL improves robustness and accuracy of models to label shift in `Imbalanced-CIFAR`. We present results across five shift severities for $\sigma = 0.1$. We observe that for all shifts, $\Delta_G > 0$, indicating that each SL model is more robust and accurate than ERM in the presence of label shifts. We observe improvements in both robustness and accuracy for the DRO CVaR algorithm, indicating that there is no tradeoff between the two.

| Dataset | ERM train perf | ERM test perf | SL train perf | SL test perf | Performance Gain | $\mathbf{\Delta_G}$ |
|---|---|---|---|---|---|---|
| MIMIC-III Notes | $0.840 \pm 0.008$ | $0.814 \pm 0.039$ | $0.827 \pm 0.000$ | $0.843 \pm 0.035$ | $\mathbf{0.029 \pm 0.004}$ | $\mathbf{0.042 \pm 0.008}$ |
| PovertyMap | $0.626 \pm 0.152$ | $0.489 \pm 0.138$ | $0.727 \pm 0.085$ | $0.602 \pm 0.090$ | $\mathbf{0.113 \pm 0.046}$ | $\mathbf{0.008 \pm 0.004}$ |

Table 6: SL improves robustness and performance of models in the presence of natural subpopulation shifts. Here we use $\sigma = 0.1$ for `MIMIC-III` and $\sigma = 0.001$ for `PovertyMap`.

Natural shifts are more difficult to accommodate than synthetic shifts because they are usually composed of multiple shifts. It is more difficult to train models to be robust against combinations of shifts because most methods are developed to deal with a single type of shift. Given that natural shifts are much harder than synthetic shifts we investigate why it is that SL is more accurate and robust. We identify that the datasets we considered contain a combination of shifts which make up the natural shift. Both `MIMIC-III` and `PovertyMap` contained both subpopulation and label shift. Thus, we believe that the improvement in performance and robustness is in part due SL being a much better learning algorithm for dealing with label shift.

# 7  CONSISTENCY OF STABLE LEARNING

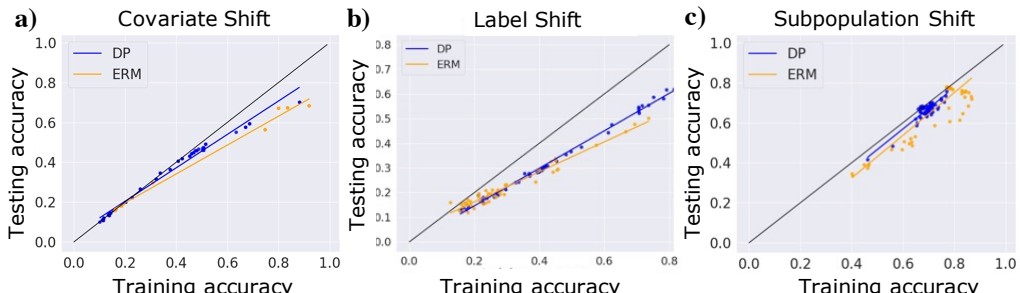

Figure 1: Stability is consistent across hyperparameters. We plot the training vs. testing accuracy across 3 representative examples of a) covariate, b) label, and c) subpopulation shift from the `CIFAR10-C`, `Imbalanced-CIFAR`, and `Waterbirds` datasets, respectively. Each point in the graph represents a different hyperparameter experiment for the dataset. SL follows the $y = x$ line more closely than ERM, indicating that the generalization gap of SL is lower than ERM and consistent across all hyperparameters.

In this section, we investigate answers to query (ii) in Section 4. We explore to what level of stability is needed and how consistent the results of the above two sections are across different model architectures and hyperparameters.

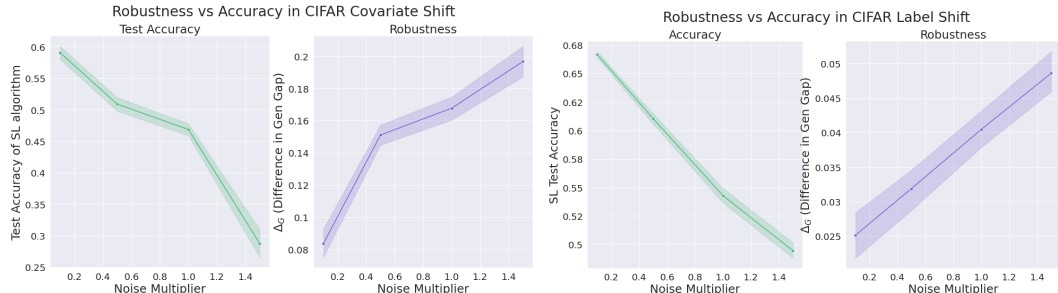

(a) Accuracy and $\Delta_G$ of covariate shift across stability levels

(b) Accuracy and $\Delta_G$ of label shift across stability levels

Figure 2: Robustness to distribution shift and accuracy are at odds for covariate shift and label shift when we use stable learning for models trained on `CIFAR-C` and `Imbalanced-CIFAR` respectively. This tradeoff worsens as the level of stability is increased.

### 7.1 LOW LEVELS OF STABILITY ARE NEEDED TO IMPROVE ROBUSTNESS

Overall, low levels of stability are required to see improvements over ERM. From Table 8, Table 11 and Fig. 2 we observe that while increasing stability leads to a lower generalization gap, it decreases performance. Over all settings, we found that $\sigma < 0.5$ best balances accuracy-robustness tradeoffs. We find that for larger values because stability is guaranteed by use of noise in DP-SGD this results in much worse accuracy. In practice, the amount of stability needed to balance this tradeoff is model and dataset dependent. As such, stability can be treated as a hyperparameter to be tuned for robustness (similar to regularization), rather than a guaranteed solution.

### 7.2 BENEFITS OF STABLE LEARNING ARE CONSISTENT ACROSS ARCHITECTURES AND HYPERPARAMETERS

In our experiments we consistently observed lower generalization gaps across all hyperparameter settings of SL models. This indicates that the robustness improvements provided by SL hold across different model settings, and are not simply a result of well-chosen hyperparameters. In Fig. 1, we examine three representative covariate, label, and subpopulation shifts. We find that the SL models more closely follow the ideal generalization trendline (in black) where the train set performance is equal to the test set performance.

Additionally, we find that our results hold across a variety of commonly used model architectures. Specifically, we use a variety of CNNs and logistic regression models across our tasks and find that the findings do not change based on the model architecture. This is expected since algorithmic stability is agnostic to architecture and hyperparameter choices by definition.

## 8 CONCLUSION

Our study investigates the utility of stability as a tool for improving both robustness and accuracy to different distribution shifts. We find that by design, stability improves robustness at the expense of accuracy for both covariate shift and subpopulation shift. Meanwhile, also by design, stability improves both robustness and accuracy to label and natural shifts. We determine that this is because of the equal importance that uniform stability places on every data point in the training set. Finally, we show that these results are consistent across hyperparamters, model architectures, and shift severities.

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
