# OpenReview forum: "Limits of Algorithmic Stability for Distributional Generalization"
_ICLR.cc/2023/Conference — Submitted to ICLR 2023_

### Official Review · Reviewer_iJTD · 2022-10-25

**Confidence:** 2
**Clarity, Quality, Novelty And Reproducibility:** Please see above
**Correctness:** 2
**Technical Novelty And Significance:** 2
**Empirical Novelty And Significance:** 2
**Recommendation:** 3

**Strength And Weaknesses:**

Strengths:
- The authors have conducted lots of experiments on various datasets and distribution shift types
- The experimental details are well documented

Weaknesses:
- The paper is lacking a message and is unclear what the conclusion from these experiments should be. The authors makes some attempts to justify the use of SL methods for robustness based on their results, but (a) there is quite large variability in the findings (eg whether SL improves both robustness and accuracy, (b) the standard deviations are large in many occasions making the comparisons even more obscure.
- Overall, I am concerned about the impact of this work given the insufficient justification and explanation of experimental results. What is the reason for different tradeoffs between robustness and accuracy observed for different types of shift? Is it because of the use of the chosen evaluation metric or something else?
- There is too much variation in the generalization gap for different shift severity levels. For example, for the Imbalanced-CIFAR, it seems strange that at shift severity level 3, the generalization gap is one order of magnitude smaller than others. And for most of them, the standard deviation is large.
- The experiment results are not conclusive and consistent across datasets. For example,  for label shift with Imbalanced-CIFAR, it seems that there is some improvement in both accuracy and generalization gap of DP-SGD over ERM. However, the improvement in accuracy is not observed when covariate and subpopulation shifts are considered. Again, for natural datasets, no concrete conclusions can be drawn as different shift severity gives different results.
- The test accuracy on CIFAR appears far from the current state of art. Why not use a larger model to evaluate all metrics?
- It is not made clear what the novelty is compared to Kulynych et al. 2020,2022?

**Summary Of The Paper:**

This paper investigates applying differentially private stochastic gradient descent (DP-SGD) for training models towards achieving high robustness against distributional shift. Experiments are done on datasets with covariate, label, or subpopulation shift using various noise multipliers in DP-SGD. The resulting performance is compared with models trained with empirical risk minimization (ERM) in terms of test accuracy and generalization gap.

**Summary Of The Review:**

I am familiar with DP-SGD and algorithmic stability, but I have not followed (and did not have time to do so during the short review period) the closely related works by Kulynych et al.. Hence, I might be misunderstanding the contributions.

The title of the paper is "limits of algorithmic stability for distributional generalization." After reading the paper, I am not sure I got an answer to what those limits are.

---

### Official Review · Reviewer_JhgW · 2022-10-25

**Confidence:** 2
**Correctness:** 2
**Technical Novelty And Significance:** 3
**Empirical Novelty And Significance:** 3
**Recommendation:** 6

**Clarity, Quality, Novelty And Reproducibility:**

The paper is very clearly written and easy to follow. May be the authors can add a word on how DP-SGD works for completeness.
See section below for other comment.

**Strength And Weaknesses:**

The topic of the paper is very interesting and highly relevant for the AI/ML community. It proposes well-chosen numerically experiments to illustrate the main claim. However, I believe that the questions being tackled cannot be answered robustly without a very large sets of experiments, varying hyperparameter, significantly larger sets of data etc. Or propose a solid theoretical analysis.

**Summary Of The Paper:**

This paper praises the training of learning algorithms by incorporating differential privacy techniques. In doing so, the idea is to be able to explicitly control the level of stability in order to be more efficient against perturbations of the data distribution. The results are essentially experimental, through a handful of real database and distribution shift settings.

**Summary Of The Review:**

“Unfortunately these works don’t provide any guidance on practical tradeoffs with robustness, except for a small positive result in Suriyakumar et al. (2021).” It seems a bit rude to qualify such a result as “small”.
“current theory provides no insight into the relationship between the level of stability and other model performance metrics, such as accuracy”. This seems a bit strong, most of the work on algorithmic stability, if not all, explicitly discuss the relation between stability and generalization (and so accuracy).

The authors raise some key questions in section 4. For example, what level of stability is needed to observe a positive difference between ERM and learning with stability.

However, I do not see any clear, unambiguous, quantitative answers to these questions. This is the main criticism of the previous contributions. This seems to depend, of course, on the model used, the current data, the settings of the hyperparameters etc... At most, three algorithms have been tested (CNN, LR, ResNet18).
The title seems to be too strong for the content of the article. No theoretical results on the limits of stability are presented, nor a vast and diverse numerical experimentation to support their claim.

Typos:
- "was was" in section 4.2
- "it is important" in section 7
- "the the" in section 4.1

---

### Official Review · Reviewer_weZZ · 2022-10-27

**Confidence:** 5
**Correctness:** 3
**Technical Novelty And Significance:** 2
**Empirical Novelty And Significance:** 3
**Recommendation:** 8

**Clarity, Quality, Novelty And Reproducibility:**

(Clarity) This work is well presented.

(Quality) High quality work, makes important contributions.

(Novelty) Novel.

(Reproducibility) Good.

**Details Of Ethics Concerns:**

N/A.

**Strength And Weaknesses:**

Strength:

1. The connection between algorithmic stability and model generalization is an interesting topic and has not been extensively studied in the context of distribution shifts empirically.

2. This paper studies three types of distribution shifts, which extends previous work [KYY+2022] and cover a wide range of interesting and practical distribution shifts in practice.

3. The finding on the effectiveness of stable training boost generalization performance under certain distribution shifts (label shifts and natural shifts) is very interesting, which is practically interesting.

Weakness:

1. [minor] The related work section can be further improved, for example, including more related work on the 'robustness to distribution shift' part.

Typos:

1. Typo in Eq. (4), should be $D^{\prime} \sim P$.


[KYY+2022] Bogdan Kulynych, Yao-Yuan Yang, Yaodong Yu, Jarosław Błasiok, and Preetum Nakkiran. What you see is what you get: Distributional generalization for algorithm design in deep learning. arXiv preprint arXiv:2204.03230, 2022.

**Summary Of The Paper:**

This paper empirically investigates the distributional generalization in the context of distribution shift. More specifically, the authors studies the performance of models trained with differentially private stochastic gradient descent (DP-SGD) under different types of distribution shifts---including covariate shift, label shift, and subpopulation shift. Through extensive experiments, the authors identify three interesting findings about stable training: (1). models trained with DP-SGD achieve smaller generalization gap across different types of distribution shifts; (2). Stable training (i.e., DP-SGD training) can serve as an effective way to boost model performance (test accuracy) under certain distribution shifts; (3). Under distribution shifts, there exists a trade-off between stability, model generalization gap, and model performance.

**Summary Of The Review:**

This paper studies an important problem in distributional generalization under distribution shifts and obtains interesting and strong empirical results, I would recommend acceptance.

---

### Official Review · Reviewer_6xKr · 2022-10-31

**Confidence:** 3
**Correctness:** 2
**Technical Novelty And Significance:** 2
**Empirical Novelty And Significance:** 2
**Recommendation:** 3

**Clarity, Quality, Novelty And Reproducibility:**

In this section I want to point out some aspects of the authors' exposition that makes it hard to follow and/or understand their findings.

- For a paper that exhaustively uses DP-SGD as its training method and discusses the effects of its hyperparameter, I am surprised to see it never being introduced explicitly.
- What is the difference between M and \mathcal{M}?
- How is the Hamming distance defined in this context? (please explicitly state)
- Should not Definition 1 be expressed for each subset of \Theta instead of \Theta? If not, and if \Theta = Im(M), then is not Pr[M(D) \in \Theta] = 1?
- Neither total variation distance, nor what T, P, and Q are supposed to stand for are introduced.
- Pg.4 What does a distribution being equal to Cartesian product of features and labels mean?
- How does covariate shift in MRI images not change disease prevalence?
- "features themselves stay the same" is confusing, something to the effect of "class-conditional densities are constant" should be used instead.
- Subpopulation shift is not sufficiently described.
- Do you compute Difference in Generalization Gap as stated or do you use an estimate of it in experiments?
- Why are different metrics used in MIMIC-III and PovertyMap?

Typos and organization:
- Please alphabetically order the references.
- Pg. 2 "algoirhtm"
- Pg. 9 "it important"


**Strength And Weaknesses:**

The main strength of the paper is that it investigates a potential mitigating algorithmic approach to an important problem; robustness under distribution shift. Although the paper does not have any novel theoretical contributions, I think papers that exhaustively and empirically investigate the implications of novel theoretical output in the field are important contributions, and as such should be encouraged.

However I do not believe this paper clears the bar in terms of making a thorough exploration of the questions at hand. The paper examines the implications of research by Kulynych et al. 2022 in three specific forms of distribution shift. First main weakness of the paper is the description of the problem setting. The authors' exposition is confusing rather than clarifying, and I will go more into the details of this in the next section.

My other main concern is that the authors' investigation falls short of answering the questions posed by their empirical results. Given the paper makes no theoretical or methodological contributions, it would be fair to expect more investigation - at least speculation - regarding the implications of their findings. Some examples of unfollowed leads are:

- That the most severe shifts favor ERM is very important finding in this context, that should have been investigated more - the authors' conclusion "stability has a limit" falls short of answering the questions that are provoked by this.
- Why might the results for synthetic and real distribution shift datasets differ so much?
- Why do we not observe a robustness - accuracy trade-off consistently?
- Why should differentially private training confer more robustness to distribution shifts vs. distributionally robust optimization?

When I started reading this paper, I thought these would be the exact types of questions investigated by the paper - in absence of this I find the paper's publication hard to justify.

**Summary Of The Paper:**

The paper experimentally investigates the effect of differentially private stochastic gradient training on robustness against three types of distribution shifts. The experiments involve data with synthetically generated distribution shifts as well as natural ones.

**Summary Of The Review:**

I believe that the current paper attempts a worthy exploration of an important subject, however it neither states the problem sufficiently clearly nor explore the implications of its results thoroughly.

---

### Decision · Program_Chairs · 2023-01-20

**Decision:**

Reject

**Justification For Why Not Higher Score:**

The paper is lacking a message and is inconclusive. The paper provides no new idea or theory and only provides an investigation into an existing method, thus more conclusive results and clear message are expected.

**Justification For Why Not Lower Score:**

N/A

**Metareview: Summary, Strengths And Weaknesses:**

The paper experimentally demonstrates that the more stable a learning algorithm is, the more robust the learned model is to co-variate, label and subpopulation shifts, at the expense of accuracy. The paper considers synthetic and natural distribution shifts, and considers Differentially Private SGD as a training algorithm. The paper builds on a previous work by Kulynych et al. 2022, and provides an relatively extensive evaluation of differentially private SGD training for achieving (distributional) robustness.

The paper's main strength is that it investigates a potentially effective approach to mitigate distribution shifts, which is important, and while it provides not major new method or theory, it evaluate a previously proposed method, differentially private SGD training, more extensively than done in the original work by Kulynych et al.
The paper's main weakness is that the paper's message and experiments are inconclusive.

Three out of the four reviewers recommend rejecting the paper. I agree with the reviewers (in particular R4) who argue that the paper is lacking a message, that the conclusions from the experiments are unclear, and that the results are not conclusive and consistent.


**Summary Of Ac-Reviewer Meeting:**

R1 recommends reject (3), and notes that the question studied is important, but finds that the problem is not stated sufficiently clearly and the implications are not explained thoroughly.

R2 recommends accept (8) and writes that: ``Through extensive experiments, the authors identify three interesting findings about stable training: (1). models trained with DP-SGD achieve smaller generalization gap across different types of distribution shifts; (2). Stable training (i.e., DP-SGD training) can serve as an effective way to boost model performance (test accuracy) under certain distribution shifts; (3). Under distribution shifts, there exists a trade-off between stability, model generalization gap, and model performance.''

R3 (borderline reject): Finds the topic interesting and important, but finds the results not sufficiently conclusive, in that the numerical results are not diverse enough. No concrete statements on why they are not sufficient or what exactly is missing, however.

R4 (reject, 3): Argues that the paper is lacking a message and that the conclusions from the experiments are unclear. There is a large variability in the findings, the standard variations are large, and the results are not conclusive and consistent.

Based on my own reading of the paper and the reviews, in particular R4's review, I also find that the paper is lacking a message and is inconclusive.